# Remote Sensing of Global Sea Surface pH Based on Massive Underway Data and Machine Learning

Zhiting Jiang [1], Zigeng Song [2,3], Yan Bai [1,2,4,*], Xianqiang He [1,2,5], Shujie Yu [2,5], Siqi Zhang [2] and Fang Gong [2,6]

1   School of Oceanography, Shanghai Jiao Tong University, Shanghai 200030, China;
    zhitingjiang@sjtu.edu.cn (Z.J.); hexianqiang@sio.org.cn (X.H.)
2   State Key Laboratory of Satellite Ocean Environment Dynamics, Second Institute of Oceanography,
    Ministry of Natural Resources, Hangzhou 310012, China; 190811080001@hhu.edu.cn (Z.S.);
    yushujie@sio.org.cn (S.Y.); zhangsiqi181@mails.ucas.ac.cn (S.Z.); gongfang@sio.org.cn (F.G.)
3   College of Oceanography, Hohai University, Nanjing 210098, China
4   Southern Marine Science and Engineering Guangdong Laboratory (Guangzhou), Guangzhou 511458, China
5   Ocean College, Zhejiang University, Zhoushan 316000, China
6   National Earth System Science Data Center, Beijing 100101, China
*   Correspondence: baiyan@sio.org.cn

**Abstract:** Seawater pH is a direct proxy of ocean acidification, and monitoring the global pH distribution and long-term series changes is critical to understanding the changes and responses of the marine ecology and environment under climate change. Owing to the lack of sufficient global-scale pH data and the complex relationship between seawater pH and related environmental variables, generating time-series products of satellite-derived global sea surface pH poses a great challenge. In this study, we solved the problem of the lack of sufficient data for pH algorithm development by using the massive underway sea surface carbon dioxide partial pressure ($pCO_2$) dataset to structure a large data volume of near *in situ* pH based on carbonate calculation between underway $pCO_2$ and calculated total alkalinity from sea surface salinity and relevant parameters. The remote sensing inversion model of pH was then constructed through this massive pH training dataset and machine learning methods. After several tests of machine learning methods and groups of input parameters, we chose the random forest model with longitude, latitude, sea surface temperature (SST), chlorophyll a (Chla), and Mixed layer depth (MLD) as model inputs with the best performance of correlation coefficient ($R^2 = 0.96$) and root mean squared error (RMSE = 0.008) in the training set and $R^2 = 0.83$ (RMSE = 0.017) in the testing set. The sensitivity analysis of the error variation induced by the uncertainty of SST and Chla (SST $\leq \pm 0.5\,^{\circ}$C and Chla $\leq \pm 20\%$; $RMSE_{SST} \leq 0.011$ and $RMSE_{Chla} \leq 0.009$) indicated that our sea surface pH model had good robustness. Monthly average global sea surface pH products from 2004 to 2019 with a spatial resolution of $0.25^{\circ} \times 0.25^{\circ}$ were produced based on the satellite-derived SST and Chla products and modeled MLD dataset. The pH model and products were validated using another independent station-measured pH dataset from the Global Ocean Data Analysis Project (GLODAP), showing good performance. With the time-series pH products, refined interannual variability and seasonal variability were presented, and trends of pH decline were found globally. Our study provides a new method of directly using remote sensing to invert pH instead of indirect calculation based on the construction of massive underway calculated pH data, which would be made useful by comparing it with satellite-derived $pCO_2$ products to understand the carbonate system change and the ocean ecological environments responding to the global change.

**Keywords:** random forest model; global sea surface pH; remote sensing inversion; total alkalinity; carbonate system

## 1. Introduction

Due to human fossil fuel combustion and deforestation, excess carbon dioxide is released into the atmosphere, and about a third of this is absorbed by the oceans [1,2].

In 2019 alone, the ocean $CO_2$ sink was $2.6 \pm 0.6$ gigatons of carbon (GtC) $year^{-1}$, fossil $CO_2$ emissions were $9.9 \pm 0.5$ GtC $year^{-1}$, and emissions from land-use change were $1.8 \pm 0.7$ GtC $year^{-1}$ [3]. However, the uptake of anthropogenic $CO_2$ emissions by seawater also reduces the pH and carbonate ion concentration of seawater, and the latter also reduces the calcium carbonate saturation states. This process is known as ocean acidification [4,5]. Changes in seawater pH can affect biochemical reactions, equilibrium conditions, and biological activity in the ocean [6] and have a dramatic effect on marine animals, especially shell-forming animals, which will have more difficulty obtaining calcium ions from seawater when its pH decreases [7,8]. It has been reported that changes in seawater pH significantly affect the growth rate of phytoplankton and consequently the primary production of phytoplankton [9–12]. Jiang et al. [13] inverted the sea surface pH distribution in 1770 by assuming that the historical rates of change of carbon dioxide partial pressure ($p$CO_2$) and sea surface temperature (SST) were consistent with the ESM2M model's calculations, and their study revealed that the average pH of seawater decreased by ~0.1 between the 17th century and the 20th century. According to the United Nations Intergovernmental Panel on Climate Change (IPCC [14]) Representative Concentration Pathway 8.5, ocean pH is expected to decrease by 0.3 in 2100 compared to 2000 [15]. Therefore, the study of ocean acidification has become a critical topic in ocean and global change research. Moreover, the study of pH, as a direct indicator of ocean acidification, is important to understand changes in ocean acidification and their underlying mechanisms.

pH is defined as the negative logarithm of the hydrogen ion concentration [6]. In current oceanographic studies, seawater pH data are mainly derived from *in situ* observations from cruise and station sampling or indirect calculations based on carbon chemistry principles [16]. On the global scale, direct measurement of pH is still far from sufficient. Because pH, $p$CO_2$, total alkalinity (TA), and dissolved inorganic carbon (DIC) are the four major carbonate system parameters, pH can be calculated from two other carbonate parameters [13,16]. Millero pointed out that the error arising from the indirect calculation of pH by using $p$CO_2$ and DIC (error $\leq \pm 0.025$), as well as by using $p$CO_2$ and TA (error $\leq \pm 0.026$), is lower compared to the calculation using TA and DIC (error $\leq \pm 0.062$) [16].

Some researchers have conducted satellite inversion of pH data to expand the spatial and temporal understanding of pH variation. The European Space Agency's Pathfinder Ocean Acidification project, led by Plymouth Marine Laboratory, UK, aims to monitor ocean acidification using Earth observations [17]. It has calculated the monthly sea surface pH distribution in the North Atlantic for 2010 using climatology $p$CO_2$ and TA, estimated from the European Space Agency's Soil Moisture and Ocean Salinity data, with an uncertainty in pH of 0.0035. The Japan Meteorological Agency released carbon dioxide mapping data (the JMA Ocean $CO_2$ Map) that contain the products of air–sea $CO_2$ flux, $p$CO_2$, TA, DIC, and pH data from 1990 to 2019 with a spatial resolution of $1° \times 1°$ and a temporal resolution of one month. For the JMA Ocean $CO_2$ dataset, the TA and DIC were produced from the model and reanalysis data based on the partition multiple linear regression method, and then the pH was indirectly calculated from the carbonate calculation based on the modeled TA and DIC data [18,19]. Copernicus Marine Environment Monitoring Service (CMEMS) also released the dataset of Global Ocean Surface Carbon Product (https://doi.org/10.48670/moi-00047. Accessed on 1 February 2022), which contains the monthly average global sea surface pH product with $1° \times 1°$ spatial resolution for 1985 to 2020; the pH products were indirectly calculated from the modeled $p$CO_2$ and reconstructed TA data based on carbonate system calculation [20]. Currently, the direct inversion of pH is still sparse.

The narrow dynamic range of pH values (~$8.1 \pm 0.1$) [16] imposes high accuracy and stability requirements on the inversion model. The control mechanism of pH change is complex, which makes pH inversion more difficult and challenging. For example, from the perspective of carbon chemistry principles, pH is strongly influenced by seawater temperature. SST influences pH by controlling the chemical speciation of $CO_2$ dissolved in seawater; however, SST also affects the air–sea exchange and the associated changes in the DIC/TA ratio [13]. These two mechanisms can cancel each other out and therefore pose

many difficulties for the inversion of pH data. Biological effects also influence the change of pH; for example, algal bloom outbreaks increase pH because of the intake of $CO_2$ in water when photosynthesis of phytoplankton is strong [21].

In the past three decades, machine learning has been increasingly used in the field of remote sensing owing to the increase in the quantity of remotely sensed data available and the development of computer technology [22]. Commonly used machine learning methods include multilayer perceptron, random forest (RF), support vector machine, and deep learning. The backpropagation (BP) neural network was more commonly used in the early days. Keiner et al. inverted the chlorophyll a (Chla) and suspended matter concentrations by using remote sensing reflectance in the first three visible bands of Landsat through a BP neural network model and found the accuracy of the model was higher than that of the conventional regression analysis [23]. Chen et al. implemented the inversion of sea surface salinity (SSS) in the Gulf of Mexico based on remote sensing reflectance and SST by constructing a BP multilayer perceptron model [24]. Breiman et al. proposed an RF algorithm based on the decision tree as a base learner to construct bagging integration with better training efficiency and lower generalization error [25]. Liu et al. constructed an RF inversion model to retrieve the PM2.5 level, based on top-of-atmosphere reflection from the Himawari satellite [26]. Chen et al. constructed a remote sensing inversion model of seawater $pCO_2$ in the Gulf of Mexico based on an RF algorithm, with SST, SSS, Chla, and diffuse attenuation of downwelling irradiance as inputs, and reported that the model had high robustness [27].

Many studies have demonstrated the unique advantages of machine learning in solving water color remote sensing inversion problems. However, the training of machine learning models still faces many challenges in terms of robustness and generalization ability owing to the lack of sufficient ground truth data, i.e., insufficient training because of few sample points in some regions.

At present, pH inversion research still faces difficulty in insufficient spatial and temporal representativeness of field-measured datasets, identifying the suitable inversion method, and the low spatial and temporal resolution of the products. In this study, we develop a new method to invert the global sea surface pH data by satellite remote sensing, based on a high-precision, large-volume pH dataset from indirect carbonate system calculations and built upon an optimized strategy of input parameter combination and machine learning. Based on the new pH inversion model, we present the temporal and spatial variation of monthly global pH from 2004 to 2019 to show model performance.

## 2. Data and Methods

### 2.1. Field Data

The field-measured data (Table 1) are mainly from the Global Surface $pCO_2$ (Lamont–Doherty Earth Observatory (LDEO)) Database V2019 [28] published by the Ocean Carbon Data System and the Global Ocean Data Analysis Project (GLODAP) dataset [29] published by the Bjerknes Climate Data Centre and the Integrated Carbon Observing System Ocean Thematic Centre.

To match the field measurements and the satellite data, we used data in the LDEO dataset from 2004 to 2019 (Figure 1), with a total data volume of 11,624,075 sets, including the underway $pCO_2$, SSS, and SST and the corresponding information of station longitude and latitude and the sampling time (year/month/day). The data used in the GLODAP dataset were from sampling stations of TA and pH above 10 m water depth from 1992 to 2019 (Figure 2); we selected the data with World Ocean Circulation Experiment (WOCE) quality control flags of 2 (=good).

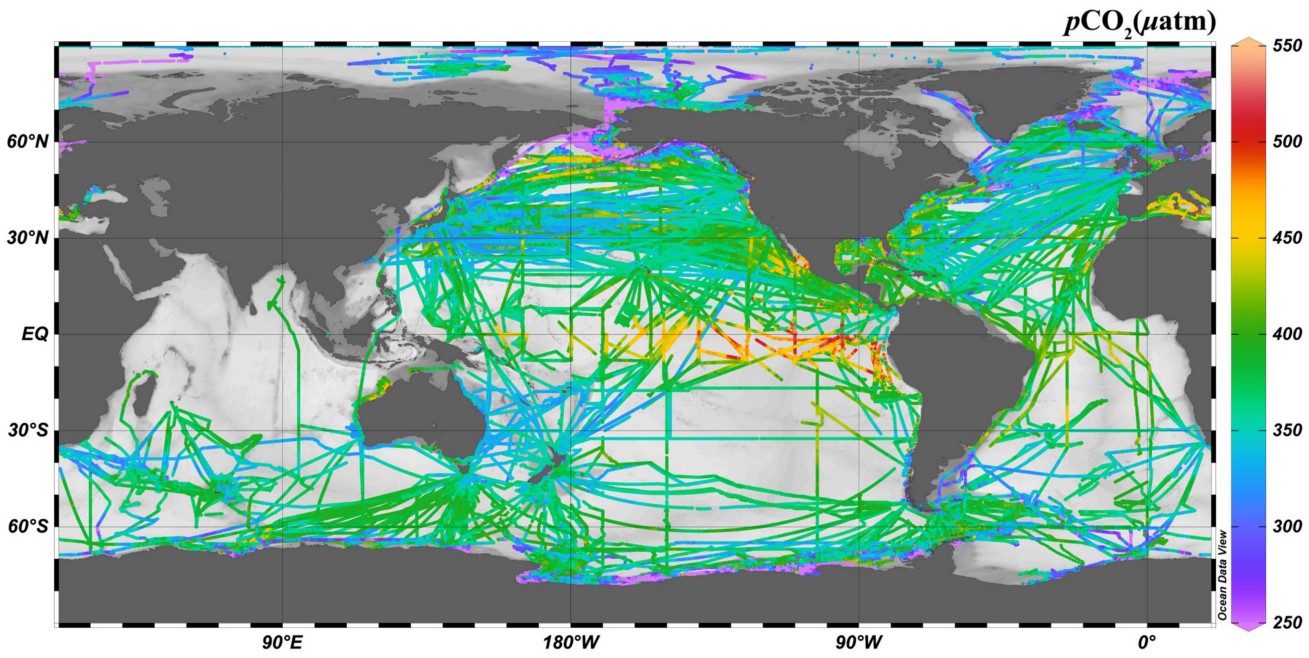

**Figure 1.** Spatial distribution of the LDEO $p$CO$_2$ dataset (2004–2019).

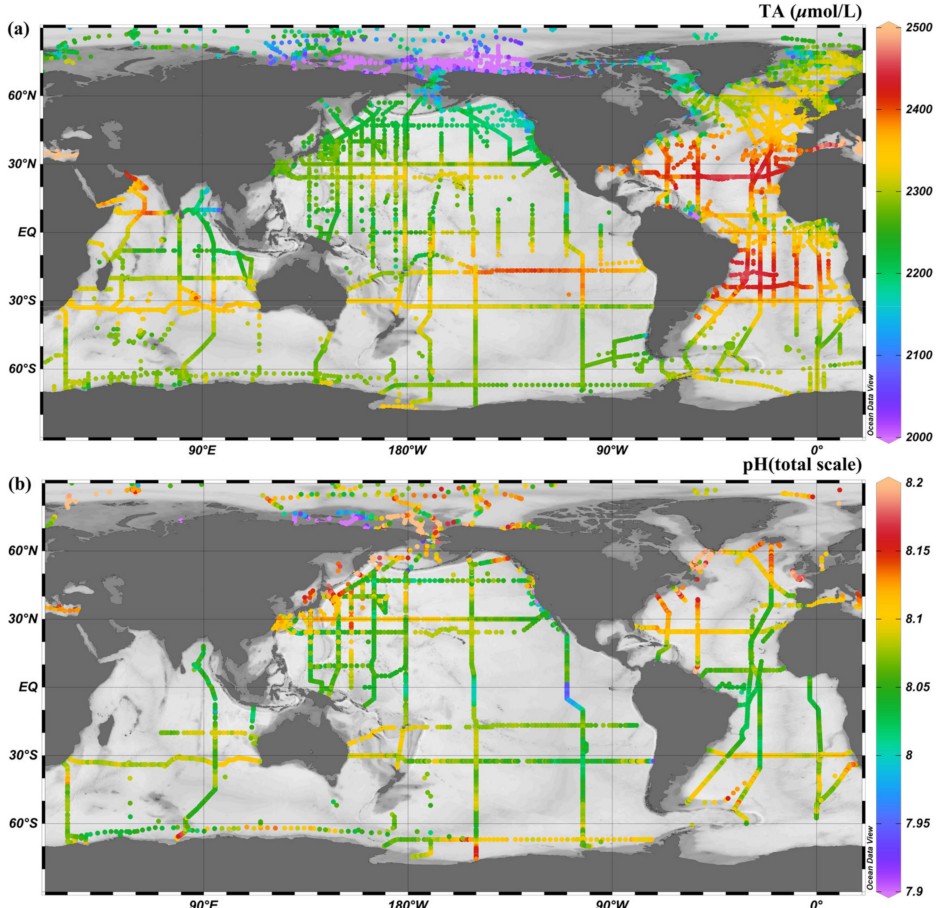

**Figure 2.** Spatial distribution of used carbonate parameters from the GLODAP dataset. *In situ* (**a**) TA and (**b**) pH variation from 1992 to 2019. The WOCE quality control flags for both TA and pH are 2 (=good).

**Table 1.** Summary information of field measurements and remote sensing data used in this study.

| Parameters | Dataset | Data Type | Time | Spatial Resolution |
|---|---|---|---|---|
| $p$CO$_2$, SST, SSS | Global Surface $p$CO$_2$ (LDEO) Database | Field data | 2004–2019 | station sampling |
| $p$CO$_2$, TA, pH, SSS, SST | GLODAP | Field data | 1992–2019 | station sampling |
| RRS, Chla | MODIS-Aqua | Satellite data | 2004–2019, daily | 4 km |
| Chla, SST | MODIS-Aqua | Satellite data | 2004–2019, monthly | 4 km |
| U-wind, V-wind | CCMP | Reanalyzed data | 2004–2019, daily | $0.25° \times 0.25°$ |
| Mixed-layer depth | CMEMS GLOBAL_MULTIYEAR_ PHY_001_030 | Reanalyzed data | 2004–2019, monthly | 4 km |

*2.2. Remote Sensing Data and Reanalysis Data*

The remote sensing data and reanalysis data used in this study were obtained from MODIS-Aqua, the Cross-Calibrated Multi-Platform (CCMP), and the CMEMS dataset for the period 2004–2019 (Table 1).

Remote sensing reflectance (RRS), Chla, and SST data at 4 km resolution at the MODIS-Aqua L3 level were downloaded from the Ocean Color website (https://oceandata.sci.gsfc.nasa.gov/. Accessed on 17 December 2020). The CCMP provided the zonal wind speed (U-wind) and meridional wind speed (V-wind) at 10 m intervals with a spatial resolution of $0.25° \times 0.25°$ and a temporal resolution of 6 h [30]. We calculated the absolute wind speed by using wind speed $= \sqrt{U - wind^2 + V - wind^2}$ and then averaged the four time intervals to obtain the daily wind speed. We also used the monthly mixed-layer depth data from GLOBAL_MULTIYEAR_PHY_001_030 dataset produced by the CMEMS with a spatial resolution of 4 km [31].

*2.3. Calculation of the Seawater Carbonate System*

The carbonate equilibrium in the ocean consists of the following components: the dissolution of $CO_2$ in aqueous solution, the formation of $H_2CO_3$ in water, the ionization of $HCO_3^-$, and the precipitation of $CO_3^{2-}$. These equilibrium relationships can be expressed as:

$$CO_2(g) = CO_2(aq) \tag{1}$$

$$CO_2(aq) + H_2O = H^+ + HCO_3^- \tag{2}$$

$$HCO_3^- = H^+ + CO_3^{2-} \tag{3}$$

$$Ca^{2+} + CO_3^{2-} = CaCO_3(s) \tag{4}$$

In addition to $p$CO$_2$, three other parameters in seawater carbonate systems are needed, namely pH, total alkalinity (TA), and dissolved inorganic carbon (DIC). These are defined as follows:

$$pH = -\log[H^+] \tag{5}$$

$$DIC = [HCO_3^-] + [CO_3^{2-}] + [CO_2] \tag{6}$$

$$TA = [HCO_3^-] + 2[CO_3^{2-}] + [B(OH)_4^-] + [OH^-] + [HPO_4^-] + 2[PO_4^{3-}]$$
$$+ [SiO(OH)_3^-] - [H^+] - [HSO_4^-] - [HF] - [H_3PO_4] \approx [HCO_3^-] + 2[CO_3^{2-}] \tag{7}$$

In Equation (7), because the proportion of $[HCO_3^-]$ and $[CO_3^{2-}]$ in TA exceeds 96%, TA can also be approximately equal to carbonate alkalinity (TA $\approx$ CA $= [HCO_3^-] + 2[CO_3^{2-}]$). If we know any two of these four parameters ($p$CO$_2$, pH, DIC, and TA), the remaining two parameters can be calculated through the carbonate calculation. In this study, we used the

MATLAB version of CO2SYS [32] (https://cdiac.ess-dive.lbl.gov/ftp/co2sys/. Accessed on 25 September 2019) to calculate the pH (total scale) at the sea temperature. The sulfuric acid dissociation constant was used as the standard, as proposed by Dickson [33]. The carbonic acid dissociation constant was adopted from the real seawater constant proposed by Lueker et al. [34], setting the seawater temperature in the range of 2–35 °C and the salinity in the range of 19–43 psu. In addition, we filtered the LDEO data according to these temperature and salinity criteria before calculating pH.

*2.4. Accuracy Evaluation Index*

As the accuracy evaluation indices of models and remote sensing products, RMSE, correlation coefficient (R), and mean bias (MB) are used in this study. RMSE measures the deviation of the model value from the field measurement data, R reflects the closeness of the linear relationship of the studied variables, and MB measures the degree to which the mean estimate of the algorithm approximates the target. The indexes were calculated as follows:

$$\text{RMSE} = \sqrt{\frac{\sum_{i=1}^{n} (x_i - y_i)^2}{n}} \tag{8}$$

$$\text{R} = \frac{\text{Cov}(X, Y)}{\sqrt{\text{Var}[x]\text{Var}[y]}} \tag{9}$$

$$\text{MB} = \frac{\sum_{i=1}^{n} (x_i - y_i)}{n} \tag{10}$$

## 3. Algorithm Development and Validation

*3.1. Overview of the pH Inversion Model Development*

Figure 3 shows the general methodological flowchart for pH inversion model development and product creation in this study. First, we constructed a global sea surface TA inversion model using the *in situ* station sampling TA data in the GLODAP dataset, and we then applied this model to the LDEO underway dataset to obtain a calculated TA for a similar scale of massive underway data. These calculated TA values were then combined with the underway $p\text{CO}_2$ data in the LDEO dataset to calculate the sea surface pH based on the carbon chemistry (Section 2.3), which is regarded as the near ground truth data (pH (*in situ*\*)). Several satellite-derived parameters were matched with the pH (*in situ*\*), which completed a large-volume dataset for further model development. The matchup dataset was randomly divided into training and testing subsets in the ratio of 7:3. Three machine learning models and multiple combinations of 11 parameters as inputs were tested, and the best-performing model was selected and validated with the LDEO dataset and the GLODAP dataset.

In the satellite-derived pH production generation session, longitude (LON), latitude (LAT), Chla, SST values from MODIS-Aqua, and MLD from CMEMS were used to produce monthly averages of sea surface pH from 2004 to 2019 with 0.25° × 0.25° spatial resolution. The satellite-derived pH data were also matched with the GLODAP *in situ* dataset for product-level validation.

The detailed processing is described in the following subsections.

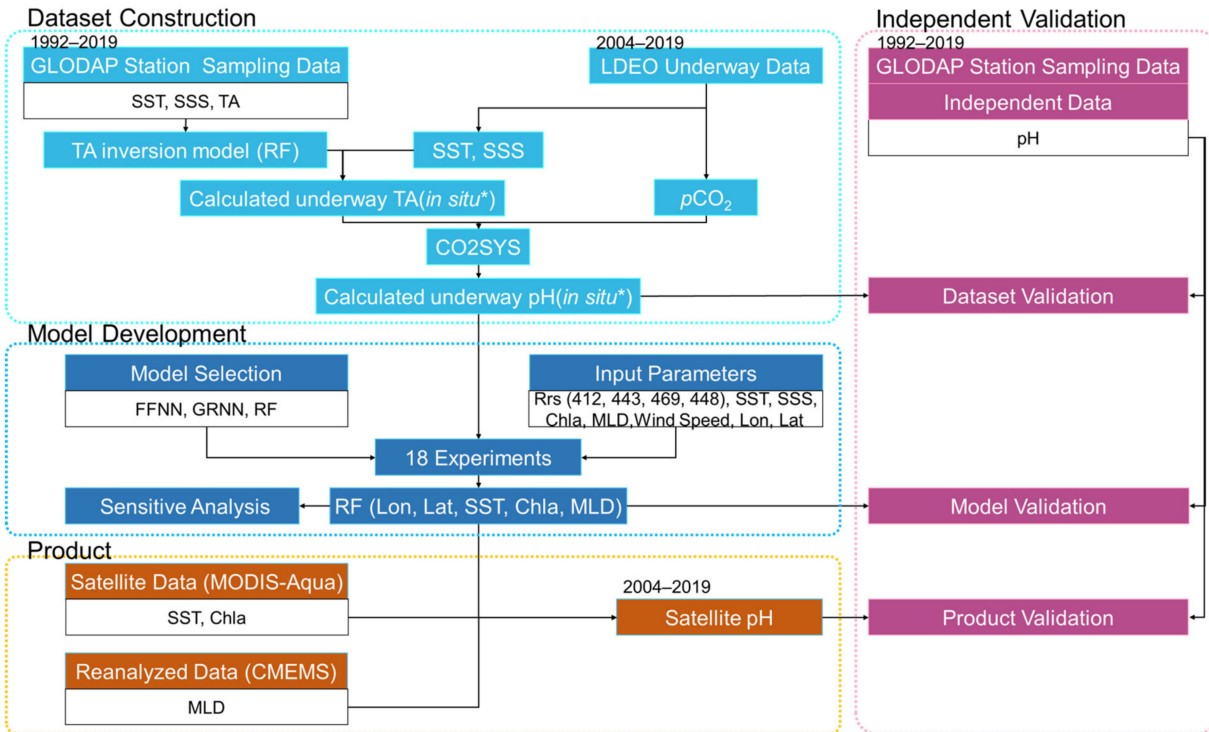

**Figure 3.** Flowchart of pH remote sensing inversion. *In situ\** means to treat the calculated data as the ground truth data.

### 3.2. Construction of the near In Situ pH Dataset

#### 3.2.1. Construction of the Calculated near *In Situ* TA Dataset

The machine-learning-based satellite inversion model requires a large amount of data for training, but the current global field-measured pH data (pH *in situ*) are quite sparse. At present, the concentrated dataset of marine biogeochemical data is from the GLODAP dataset, in which the pH data volume of sea surface data (at a sampling depth of <10 m) after 2004 comprises only 8195 sets on the global scale, which is far from satisfying the training requirements for a global model. Therefore, it is necessary to construct a spatiotemporally representative calculated pH dataset as the ground truth data for model training (pH (*in situ\**)). Because pH can be indirectly calculated from other carbonate parameters, the pH (*in situ\**) obtained from the massive underway measured global $pCO_2$ dataset (e.g., the LDEO or the Surface Ocean $CO_2$ Atlas dataset) was used to expand the data volume and improve the spatial and temporal representativeness of the sea surface pH dataset. It is generally accepted that the error in calculating pH from TA and $pCO_2$ and from DIC and $pCO_2$ is minimal [16]. Because TA reflects the ability of water bodies to absorb protons, sea surface TA is mainly controlled by freshwater addition and removal, which is similar to the control mechanism of salinity [35]. Therefore, TA is relatively stable and less affected by the action of marine organisms [36]. The calculation of the global TA based on salinity has been studied extensively [19,36–39]. An empirical relationship between salinity-normalized alkalinity (NTA = TA × 35/salinity) and the second-order polynomial of seawater temperature has been built by Millero et al. for inversion of TA [36]. Lee et al. later constructed a polynomial regression empirical formulation to estimate TA based on SSS, SST, and longitude in five major global sea areas; this has been the most widely used TA inversion algorithm [38]. Takatani et al. improved upon the work of Lee et al. by dividing the Pacific Ocean into five areas and constructing a polynomial relationship between TA and SSS and sea surface dynamic height [39]. All these studies exhibited good performance in TA inversion accuracy. In our study, TA is an intermediate value used for calculating the pH dataset. To avoid the boundary problem caused by area partitioning and to reduce

the error propagation in the calculation, our study used a machine learning approach to reconstruct the TA model based on a machine learning method and the temperature and salinity as inputs.

We compared two models, the forward feedback neural network (FFNN) and the RF model, to generate the inverted TA model. The datasets for TA model training and testing came mainly from the field-measured data in the GLODAP dataset. We selected the TA, SSS, and SST data with a WOCE quality control flag of 2 (=good) in GLODAP and a sampling depth of <10 m. We generated a matchup dataset of TA, LON, LAT, SST, and SSS, and we used the field-measured pH data from 1992 to 2019. Data from earlier years were excluded by considering that the collection and quality control methods of earlier data might be different. We used data from 1992 to 2014 as the training set and data from 2015 to 2019 as the independent validation set.

Through four sets of model training (Table 2), we found that the overall accuracy of the RF algorithm was higher compared with that of the FFNN algorithm, and the performance of the model was more stable in the testing dataset after adding LON and LAT as model inputs. Therefore, a TA inversion model based on the RF model with input parameters of LON, LAT, SST, and SSS was selected. From the error distribution of the training and testing datasets (Figure 4a,c), we found that the spatial distribution of error was uniform. From the scatter plots of the *in situ* and estimated values (Figure 4b,d), we found that the modeled TA values of both the training and testing datasets were also well distributed along the 1:1 line. The modeled errors in the Arctic Ocean were relatively high, owing to the special carbonated system and the great influence of river flushing water and glacial meltwater on SSS. In general, this TA model on a global scale can be used for further pH inversion.

**Table 2.** Training results of TA inversion models.

| | Model | Input | $n$ (Training) | Time | $R^2$ | RMSE | $n$ (Testing) | Time | $R^2$ | RMSE |
|---|---|---|---|---|---|---|---|---|---|---|
| 1 | FFNN | SST, SSS | 17,743 | 1992–2014 | 0.92 | 16.92 | 2553 | 2015–2019 | 0.97 | 12.08 |
| 2 | FFNN | LON, LAT, SST, SSS | 17,743 | 1992–2014 | 0.97 | 11.06 | 2553 | 2015–2019 | 0.98 | 9.35 |
| 3 | RF | SST, SSS | 17,743 | 1992–2014 | 0.98 | 9.39 | 2553 | 2015–2019 | 0.96 | 13.62 |
| 4 | RF | LON, LAT, SST, SSS | 17,743 | 1992–2014 | 0.99 | 11.19 | 2553 | 2015–2019 | 0.98 | 13.38 |

### 3.2.2. Construction and Validation of the Calculated near *In Situ* pH Dataset

Based on the above-mentioned TA model, we inputted the underway SST and SSS from the LDEO dataset to generate the near *in situ* TA values. We then matched them with the underway $pCO_2$ data in the LDEO dataset. Next, we calculated the pH through CO2SYS (Section 2.3) based on the TA and $pCO_2$ data. Finally, a large-volume global pH dataset of comparable size to the LDEO underway $pCO_2$ dataset was generated as the near ground truth data for model development (Figure 5).

Because the station sampling data in GLODAP was not used in the construction of the calculated pH (*in situ**) dataset, we used them to validate our method as independent data. The *in situ* pH, TA, and pCO₂ of seawater in the temperature range of 2–35 °C, salinity in the range of 19–43, and sampling depth of <10 m in the GLODAP dataset from 1992 to 2019 were extracted. We used the *in situ* temperature and salinity data as input to calculate the pH (*in situ**) and compared it with the *in situ* pH in the GLODAP dataset (Figure 6). The calculated TA (*in situ**) had RMSE = 2.82 $\mu$mol/kg, $R^2$ = 0.99, and $n$ = 560, while the calculated pH (*in situ**) had RMSE = 0.009, $R^2$ = 0.95, and $n$ = 560; therefore, both of these parameters exhibited high accuracy. Figure 6a shows the spatial distribution of the validation with overall differences within ±0.01. Figure 6b shows the point-to-point comparison, with pH mainly concentrated between 8 and 8.15. These points fall well along the 1:1 line, indicating that our calculation method can guarantee the accuracy of the pH (*in situ**) dataset and can be used as ground truth values for pH satellite inversion modeling.

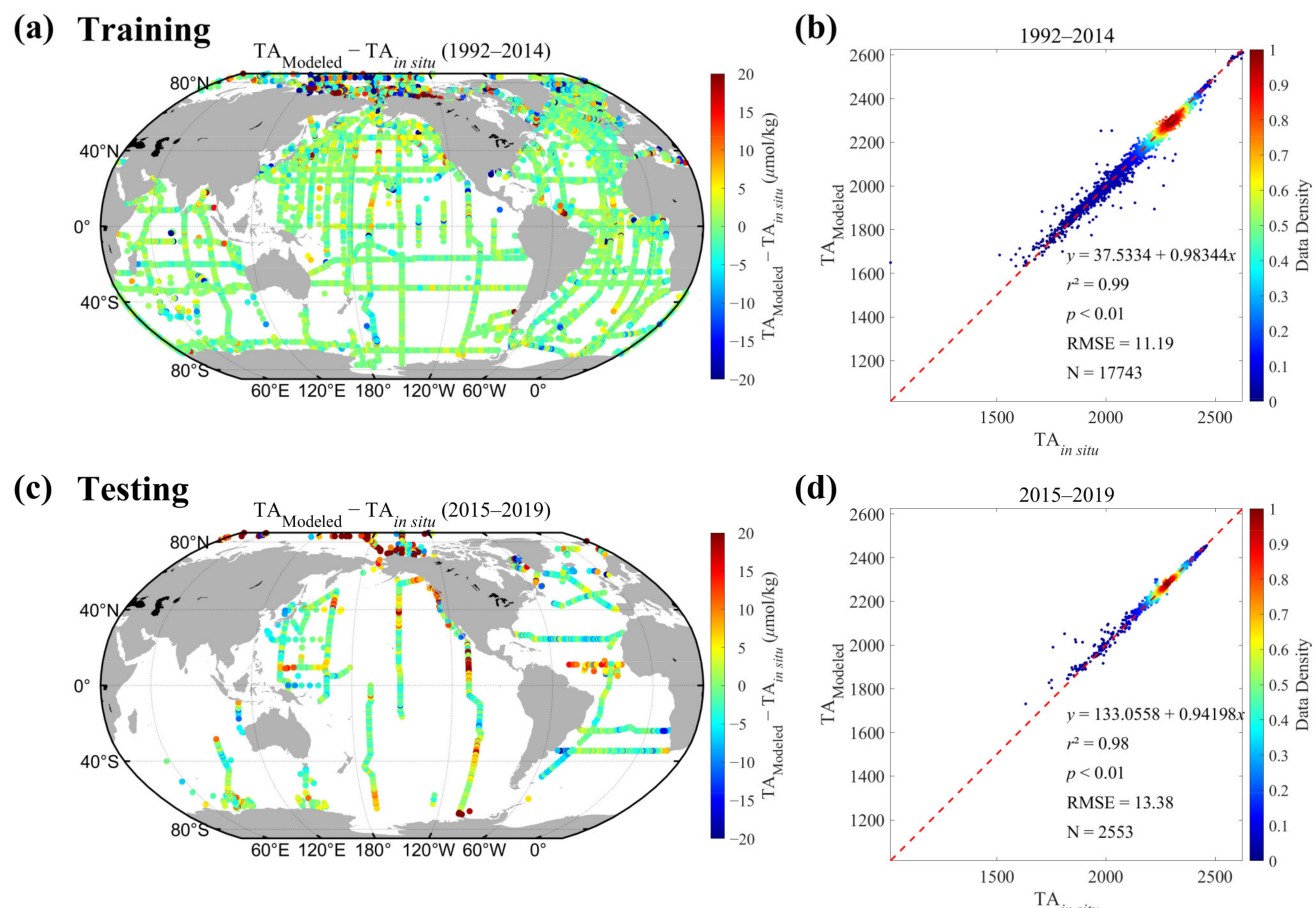

**Figure 4.** Training and testing of the TA inversion model based on the RF algorithm with inputs of LON, LAT, SST, and SSS. Spatial distributions of relative errors (the difference between the modeled value and TA (*in situ\**)) in (**a**) the training set (1992–2014) and (**c**) the testing dataset (2015–2019). Scatter plots of the comparison between the modeled value and TA (*in situ\**) for (**b**) the training set and (**d**) the testing dataset. The red dotted line is 1:1 line.

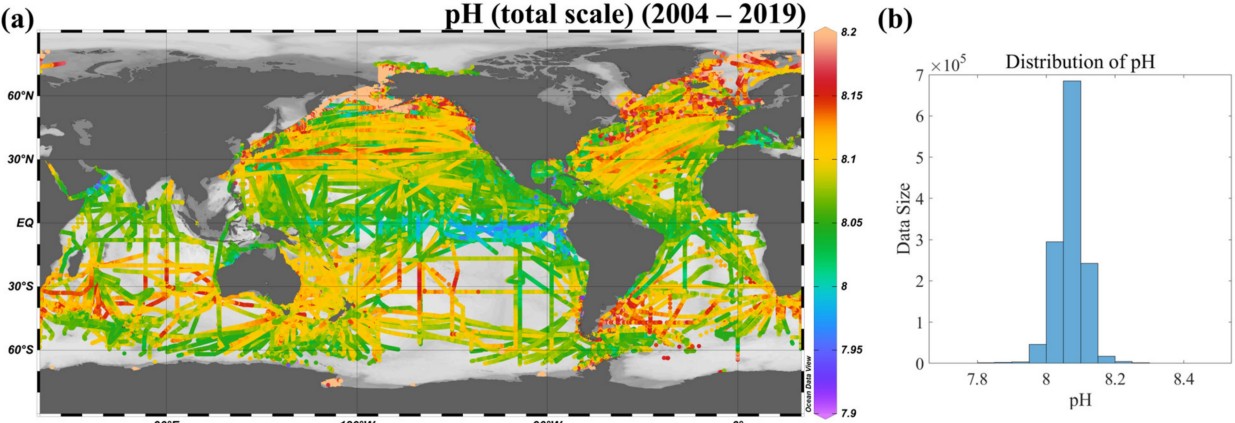

**Figure 5.** (**a**) Spatial distribution of the global pH dataset calculated from the *in situ* $p$CO$_2$ and the calculated TA. (**b**) Histogram of the pH dataset.

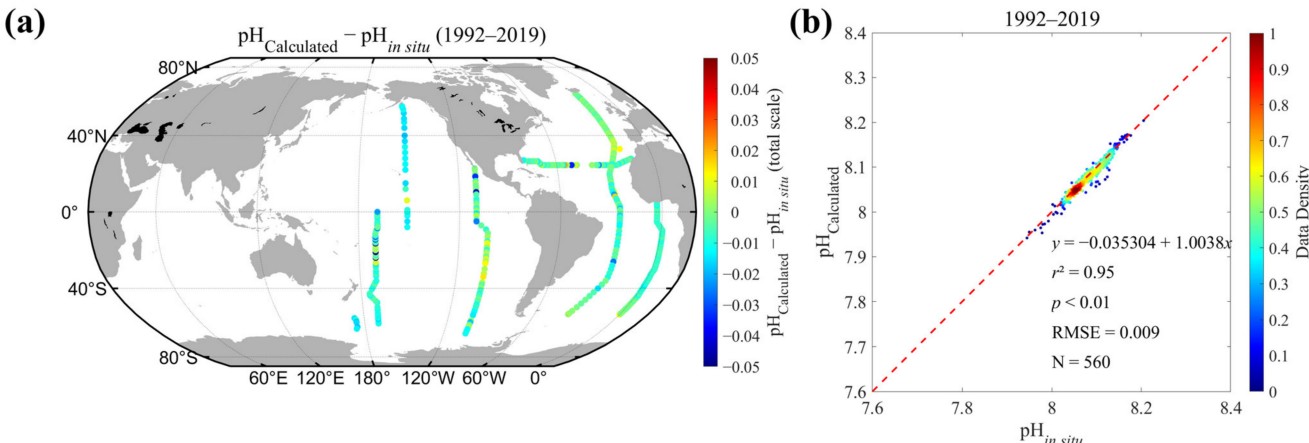

**Figure 6.** Validation of the calculated pH (*in situ\**) based on the independent GLODAP dataset. (**a**) Comparison between pH (*in situ\**) and the station sampling from GLODAP during the period 1992–2019 with data above 10 m water depth with a WOCE quality control flag of 2. (**b**) Scatter plot between the calculated pH and *in situ* pH in GLODAP along the 1:1 line with R$^2$ = 0.95, RMSE = 0.009, and *n* = 560.

### 3.3. Inversion Model of Satellite-Derived pH

### 3.3.1. Matchup Dataset for pH Inversion

The pH (*in situ\**) as the ground truth value needs to be further matched with the satellite data and reanalyzed data, and a matchup dataset with LON, LAT, YEAR, MONTH, DAY, SST, SSS, $p$CO$_2$, TA (*in situ\**), pH (*in situ\**), Chla, RRS (412, 443, 469, 488, 531, 547, 555, 645, 667, and 678 nm), mixed-layer depth, and wind speed from 2004 to 2019 was obtained for a total number of datasets of 1,302,641. The spatial distribution of the data sites is shown in Figure 7. Note that polar areas have a sparser distribution because data recorded below 2 °C were excluded in the preprocessing process in accordance with the carbonate calculation in CO2SYS.

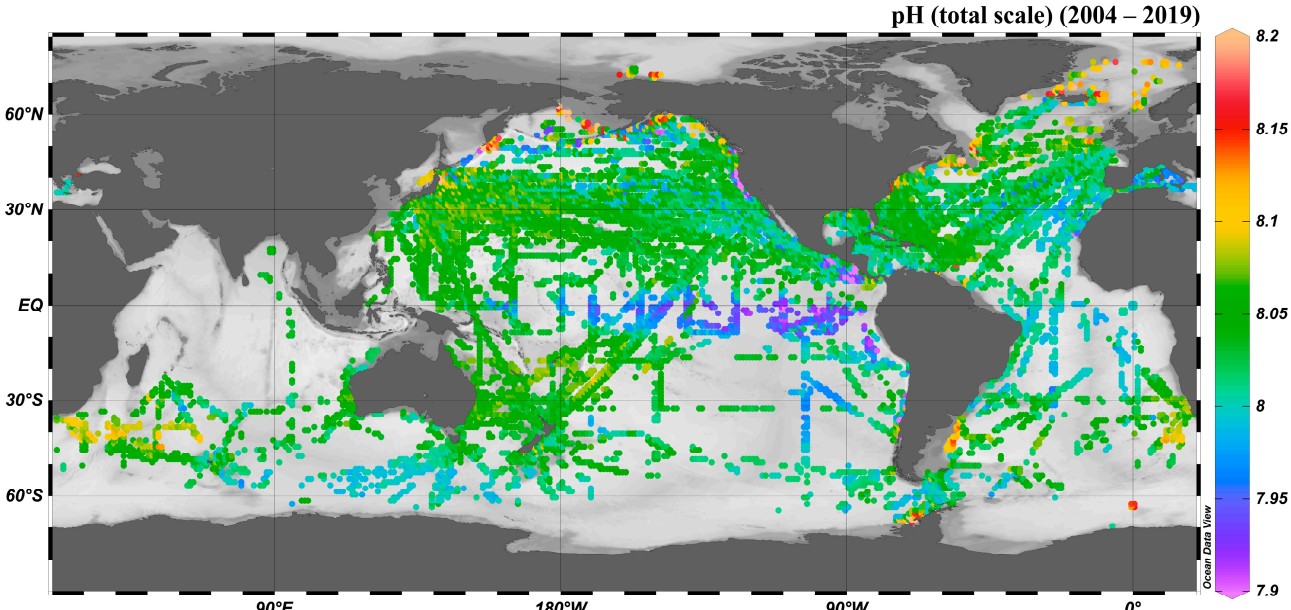

**Figure 7.** Spatial distribution of the pH (*in situ\**) dataset matched with satellite data during the period 2004–2019 with 1,302,641 sets of data.

A two-by-two correlation analysis of the parameters in the matched data (Figure 8 and Table 3) revealed that pH correlated highly with SST, RRS (412, 443, 469, and 488 nm), Chla, MLD, and wind speed. Therefore, these parameters were included as alternative input parameters for the pH satellite inversion model. From the controlling factors influencing changes in pH, the increase in SST causes an increase in the apparent ionization equilibrium constant in the carbonic acid ionization process, which in turn decreases the seawater pH. The increase in Chla somewhat enhances the photosynthetic capacity of phytoplankton in the water column, which in turn consumes $CO_2$ in the water column and increases the pH. Because salinity had a good correlation with TA, and TA is an important parameter in the carbonate calculation process of pH, we also set up a control experiment to discuss the effect of SSS on the model. In addition, salinity directly affects the ionic strength in the water column. When the salinity increases, the ionic strength increases and the apparent ionization equilibrium constant becomes smaller, which in turn causes an increase in pH. Wind speed affects water mass movement, which in turn affects water mass mixing with different carbonate system characteristics and sea–air $CO_2$ exchange, and mixed layer depth reflects the influence of deep-water masses on the surface water. The effect of wind speed and mixed layer depth on pH is also reflected in their correlation with pH ($R_{Wind\ Speed} = 0.08$, $R_{MLD} = 0.15$). Therefore, we set up experiments to illustrate the effects of wind speed and MLD on the pH inversion model. Meanwhile, we noticed that LON and LAT also have a high correlation with pH, and we added latitude and longitude as inputs to the machine learning model and set up a control group in the experiments for comparison.

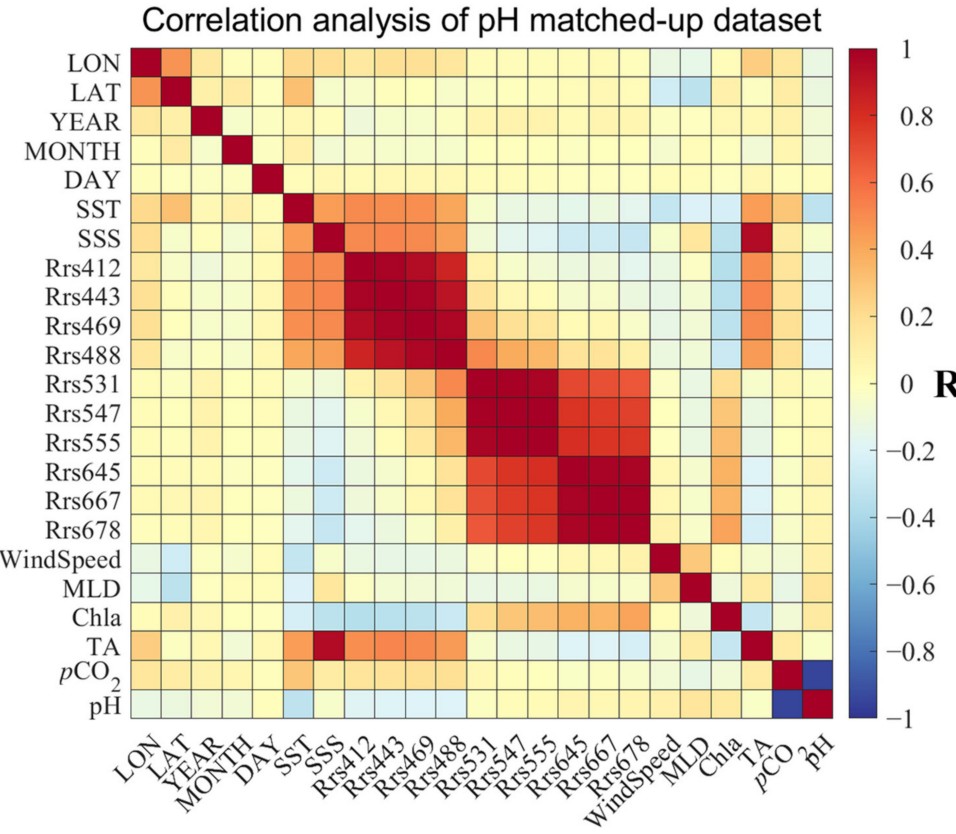

**Figure 8.** Correlation analysis matrix of pH underway dataset with related parameters. The R-values of the pH-related parameters in the matrix were all significant (*p*-value less than 0.01).

**Table 3.** The R-values of the pH-related parameters.

| | Lon | Lat | Year | Month | Day | SST | SSS | Rrs412 | Rrs443 | Rrs469 | Rrs488 |
|---|---|---|---|---|---|---|---|---|---|---|---|
| **pH** | −0.13 | −0.12 | −0.08 | −0.08 | 0.01 | −0.32 | −0.05 | −0.18 | −0.19 | −0.20 | −0.20 |
| | **Rrs531** | **Rrs547** | **Rrs555** | **Rrs645** | **Rrs667** | **Rrs678** | **Wind Speed** | **MLD** | **Chla** | **TA** | **$p$CO$_2$** |
| | −0.02 | 0.02 | 0.03 | 0.06 | 0.04 | 0.06 | 0.08 | **0.15** | **0.13** | −0.04 | **−0.95** |

### 3.3.2. Machine Learning Models and Inputs

In this study, three machine learning models were selected for model training: the classical FFNN model, the generalized regression neural network (GRNN), and the RF model. The FFNN model is a multilayer perceptron model, and we configured three implicit layers for the training of FFNN networks with different input parameters and set the number of nodes in each implicit layer separately. A total of 10 repetitions of each network structure were performed, and the average RMSE was calculated. The network with the lowest average RMSE was selected as the final network structure. For the training of GRNN networks, the main parameter that affects the model is the propagation rate of its network. We set the propagation rate between 0.1 and 2 and trained at intervals of 0.1, used 4-fold cross-validation for each training to improve the stability of the training, and finally selected the model with the lowest RMSE based on the RMSE value of each training to obtain the best propagation rate. The RF algorithm was set up with 1200 decision trees in training with a maximum tree depth of 20 and a 30-fold cross-validation of the training data.

Considering three models and six combinations of input parameters, we set up experiments with a total of 18 groups. Eleven parameter combinations were tested for the effects of the addition of latitude and longitude, the addition of salinity, MLD and wind speed, and the effects of the choice of Chla and four bands of RRS. We divided the data from 2004 to 2019 into training and testing sets by 7:3, randomly. To improve the training efficiency, and by considering continuous high-frequency underway observations (e.g., measurements taken per minute), we sorted the dataset by time and randomly sampled between every 20 data points to achieve a reduced weighting of data from the similar time and location in sampling. The final global data volume of the training set was 45,592 sets, and that of the validation set was 19,540 sets.

### 3.3.3. Model Experiments and Comparison

The training and validation accuracy results of the 18 groups of experiments are shown in Figure 9 and given in Table 4. The experimental results are shown in a polar coordinate system, with groups A–F (upper right) being the results under the FFNN model, groups G–L (lower) being the results under the GRNN model, and groups M–R (upper left) being the results under the RF model. From the perspective of the three types of models, the experiments under the FFNN model were closer to the origin in the distribution of each circle point, indicating that their $R^2$ values were lower and that the model inversion ability was weaker; the experiments under the RF model have a relatively larger $R^2$ and stronger model fitting ability. The GRNN and RF models present similar inversion capabilities, but the GRNN model is not so suitable, because it is unstable and the generated satellite products are prone to uneven color blocks. Meanwhile, the RF model-derived results have better performance.

In Table 4, the model with RRS (A, G, and M) as input has higher accuracy than the model with Chla (B, H, and N) as input due to the input of more relevant parameters. However, we found that Chla already occupies nearly half of the feature importance (Figure 10b) and has better model interpretation compared with RRS. To avoid the error propagation and information redundancy, Chla is chosen as the input of the model instead of RRS.

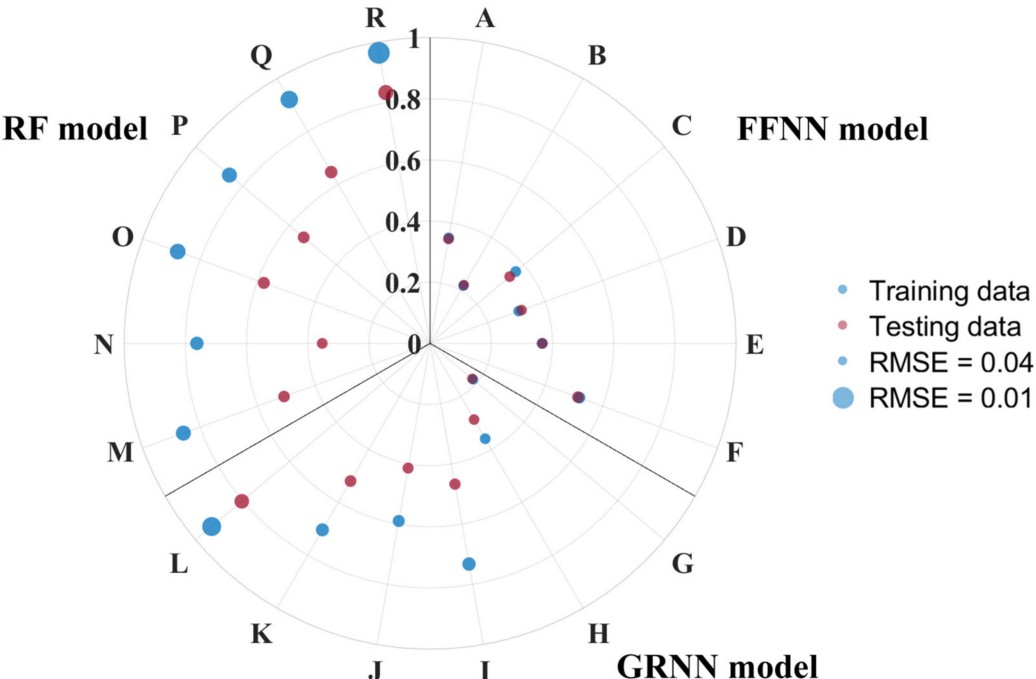

**Figure 9.** Training results of pH inversion models for 18 groups of experiments. A–F are from the FFNN model, G–L are from the GRNN model, and M–R are from the RF model. The two-colored points with the same polar angle correspond to the same set of experiments, with the blue points representing the results of the training set and the red points representing those of the testing set. The polar diameter of the dots represents the size of R²; the closer to the origin the dots are, the smaller R² is, and vice versa. The radius of the dot is taken as the reciprocal of the RMSE (i.e., 1/RMSE), and the smaller the RMSE, the larger the radius of the dot. The sizes of the dots with RMSE = 0.01 and 0.04 are also given in the figure for reference.

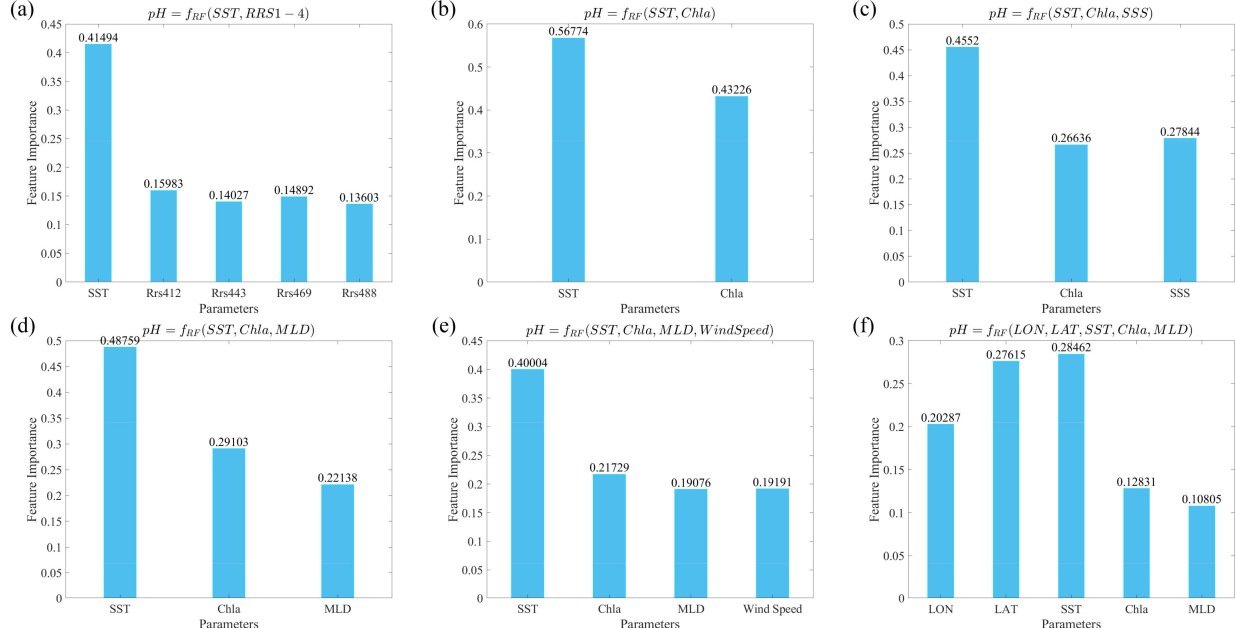

**Figure 10.** Feature Importance of sea surface pH remote sensing inversion model. (**a**–**f**) are for experiments of M–R.

**Table 4.** Training and testing results of 18 sets of pH inversion models.

| Model | Order | Parameters | Training (*n* = 45,592) | | Testing (*n* = 19,540) | |
|---|---|---|---|---|---|---|
| | | | $R^2$ | RMSE | $R^2$ | RMSE |
| FFNN | A | SST, RRS1-4 | 0.35 | 0.036 | 0.35 | 0.035 |
| | B | SST, CHL | 0.22 | 0.039 | 0.22 | 0.039 |
| | C | SST, CHL, SSS | 0.37 | 0.035 | 0.34 | 0.036 |
| | D | SST, CHL, MLD | 0.31 | 0.037 | 0.32 | 0.036 |
| | E | SST, CHL, MLD, WS | 0.37 | 0.035 | 0.37 | 0.035 |
| | F | LON, LAT, SST, CHL, MLD | 0.52 | 0.031 | 0.51 | 0.030 |
| GRNN | G | SST, RRS1-4 | 0.18 | 0.040 | 0.18 | 0.040 |
| | H | SST, CHL | 0.36 | 0.035 | 0.29 | 0.037 |
| | I | SST, CHL, SSS | 0.73 | 0.023 | 0.47 | 0.032 |
| | J | SST, CHL, MLD | 0.59 | 0.028 | 0.41 | 0.034 |
| | K | SST, CHL, MLD, WS | 0.70 | 0.024 | 0.52 | 0.030 |
| | L | LON, LAT, SST, CHL, MLD | 0.93 | 0.012 | 0.80 | 0.019 |
| RF | M | SST, RRS1-4 | 0.86 | 0.018 | 0.51 | 0.031 |
| | N | SST, CHL | 0.76 | 0.023 | 0.35 | 0.035 |
| | O | SST, CHL, SSS | 0.88 | 0.017 | 0.58 | 0.028 |
| | P | SST, CHL, MLD | 0.86 | 0.018 | 0.54 | 0.030 |
| | Q | SST, CHL, MLD, WS | 0.92 | 0.014 | 0.65 | 0.026 |
| | **R** | **LON, LAT, SST, CHL, MLD** | **0.96** | **0.009** | **0.83** | **0.018** |

Comparing the results after adding the SSS parameters (e.g., experiments N and O) shows that the addition of salinity is more helpful to the overall accuracy of the model. The feature importance of SSS is comparable to that of Chla (Figure 10c). However, considering the lack of satellite-derived salinity data over longer time spans with higher accuracy at the current stage, we did not add salinity to the final model inputs.

MLD and wind speed are also important biogeochemical parameters that affect pH. From Figure 10d,e, the inclusion of MLD and wind speed helps significantly to improve the accuracy of the model, while the feature importance of both MLD and wind speed is relatively high. However, the dynamic range of wind speed within days and months is larger compared to MLD, and wind speed and pH are not simple linear mapping. Therefore, training the model with daily-averaged wind speed products and then using monthly-averaged wind speed to produce pH products will bring more uncertainties. On balance, only the MLD was retained as an input parameter.

In multiple groups of experiments for both FFNN, GRNN, and RF, experiments (F, L, and R) with LON and LAT included in the input parameters exhibited higher $R^2$ and smaller RMSE, which corroborates the effect of adding geographic identifiers to improve the model fitting ability. The addition of the LON and LAT parameters may increase the error when inverting the pH for areas not included in the training dataset. Among the three models, the RF model has a better generalization ability and has the inversion ability for a broad area on a global scale; as we have built a massive pH (*in situ**) underway dataset that covers the majority of global ocean, we chose to include the LON and LAT as model inputs. Finally, the RF model with LON, LAT, SST, Chla, and MLD as input was determined to be the most suitable model for the sea surface pH inversion.

The comparison between the modeled results and validation data and the spatial distribution of their differences are shown in Figure 11. Figure 11a shows that the spatial distribution of the training set was relatively uniform, the spatial distribution was covered

worldwide except for the Indian Ocean, and the errors (difference between modeled results and *in situ* data) were basically ranging between ±0.01. From Figure 11c, the spatial distribution of the data in the testing set is seen to be similar to that in the training set with an error within ±0.01. From Figure 11b,d, the pH values are also seen to be basically distributed along the line of 1:1. Overall, there were no significant systematic biases present. A relatively significant error appeared in the coastal sea because the hydrological and biogeochemical environment of the marginal sea was more complex. Overall, the training accuracy of the model was satisfactory.

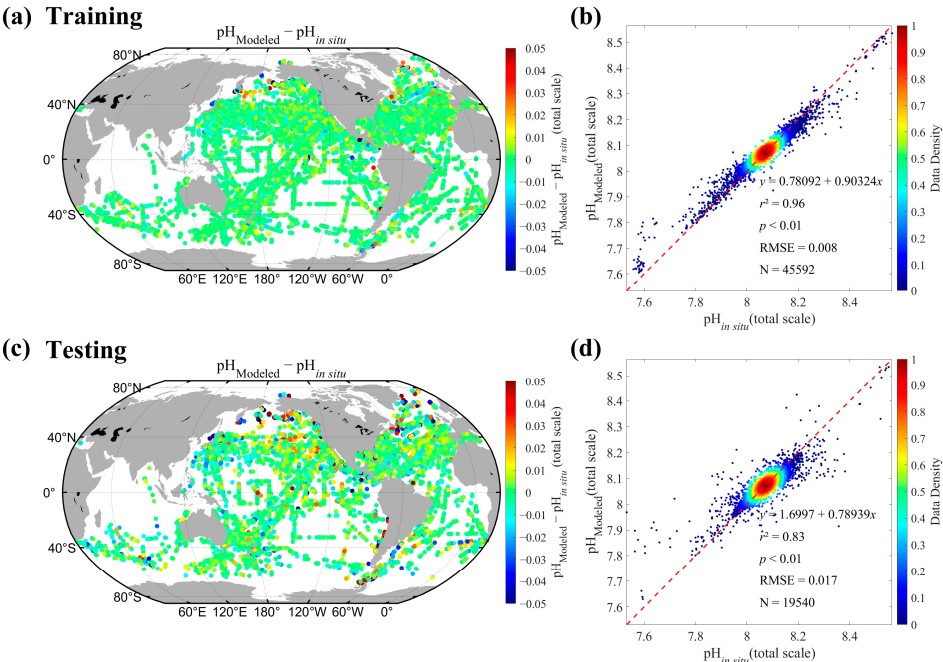

**Figure 11.** Validation of the pH inversion model based on the RF model with inputs of LON, LAT, SST, Chla, and MLD. Spatial distributions of inversion errors for the pH (**a**) training dataset and (**c**) testing dataset. Point-to-point comparison between *in situ* pH and modeled results in (**b**) the training dataset ($R^2 = 0.96$, RMSE = 0.008, and $n = 45,592$) and (**d**) the testing dataset ($R^2 = 0.83$, RMSE = 0.017, and $n = 19,540$). The red dotted line is 1:1 line.

### *3.4. Validation of the Satellite-Derived pH Product*

3.4.1. Sensitivity Analysis of the pH Model

The inputs of satellite-derived parameters inherently contain uncertainties that would generate error propagation. To test such an effect, we added some random error to the input parameters separately as the model sensitivity analysis. Gentemann reported that the standard deviation of MODIS-derived SST was 0.58 °C [40]. The uncertainty of the MODIS-derived concentration sensor was generally within ±35% [41,42]. Four scenarios of +0.5 °C, −0.5 °C, +1 °C, and −1 °C were set to the SST, and four scenarios of +20%, −20%, +35%, and −35% were set to the Chla. The results of the sensitivity analysis are shown in Figure 12 and given in Table 5. The effect of SST on pH inversion was greater than that of Chla, but outputs did not exhibit a large degree of shift overall and the model still maintained good stability. The model output values produced a negative shift when SST was elevated, which is consistent with the carbon chemistry principle: when the seawater temperature increases, the hydrogen ion activity in seawater increases and the pH value decreases. Chla had a positive effect on pH, and when Chla increased, the model MB was positive. Phytoplankton photosynthesis was expected to be stronger when Chla was high and elevating the pH of the water column. From the RMSE values in Table 5, it can be seen that the effects of different levels of SST and Chla produced errors of up to ~0.01 on

the inverted pH results. This indicates that the model has good robustness and can still maintain good accuracy overall when encountering a certain error of the input parameters.

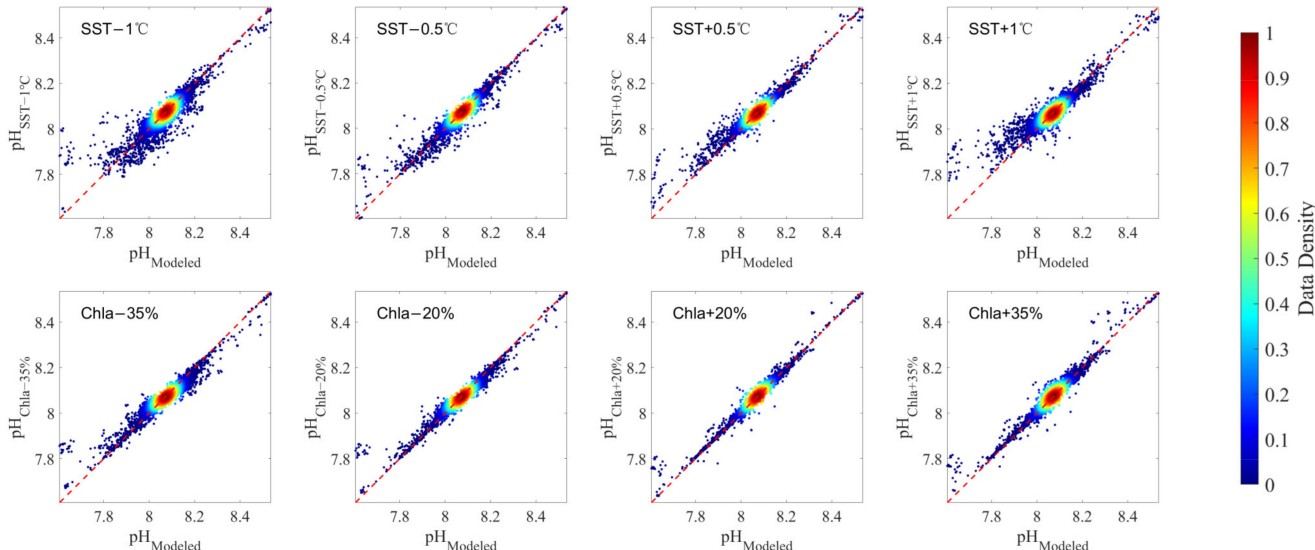

**Figure 12.** Sensitivity test of the pH inversion model based on the RF model. Uncertainties of ±0.5 °C and ±1 °C were added to the input parameter of SST, and uncertainties of ±20% and ±35% were added to the input parameter of Chla, and the stability of the output pH was compared with that of the original input. The red dotted line is 1:1 line.

**Table 5.** Results of sensitivity analysis of pH inversion model based on the RF model.

| Cases | $R^2$ | RMSE | MB | Cases | $R^2$ | RMSE | MB |
|---|---|---|---|---|---|---|---|
| SST − 1 °C | 0.94 | 0.013 | 0.0031 | Chla − 35% | 0.97 | 0.011 | −0.0004 |
| SST − 0.5 °C | 0.98 | 0.008 | 0.0016 | Chla − 20% | 0.98 | 0.007 | −0.0003 |
| SST + 0.5 °C | 0.98 | 0.008 | −0.0012 | Chla + 20% | 0.99 | 0.006 | 0.0008 |
| SST + 1 °C | 0.95 | 0.013 | −0.0025 | Chla + 35% | 0.98 | 0.008 | 0.0013 |

### 3.4.2. Validation of Satellite-Derived pH

With the above pH inversion model, we used the Chla and SST of MODIS-Aqua and MLD of CMEMS to produce monthly averages of sea surface pH from 2004 to 2019 with 0.25° × 0.25° spatial resolution. For the validation, the independent *in situ* sea surface pH data in the GLODAP dataset were used to validate both the pH inversion model and the satellite-derived pH products.

For model testing, *in situ* pH data from the GLODAP dataset from 2004 to 2019 were used as an independent dataset. The matching results were 1282 validation points (Figure 13a), which exhibited relative spatial uniformity with errors within ±0.05. From Figure 13b, the point-to-point comparison shows that the points basically fall along the 1:1 line, with $R^2$ = 0.36 and RMSE = 0.050, which also indicates stratified performance.

The validation of the satellite-derived pH product entailed verifying the product accuracy after matching with the *in situ* pH data in GLODAP from 2004 to 2019. The general accuracy of the product was maintained at a good level with $R^2$ = 0.54 and RMSE = 0.029 (Figure 14). Overall, the error of satellite-derived pH was distributed uniformly with the majority of data within ±0.03. The satellite-derived pH was slightly overestimated in the northwest Pacific Ocean near the Bering Sea, while it was slightly underestimated along the east coast of North America. Given that the satellite-derived pH was a monthly average whereas the *in situ* values were sampled instantly, such inversion results are satisfactory.

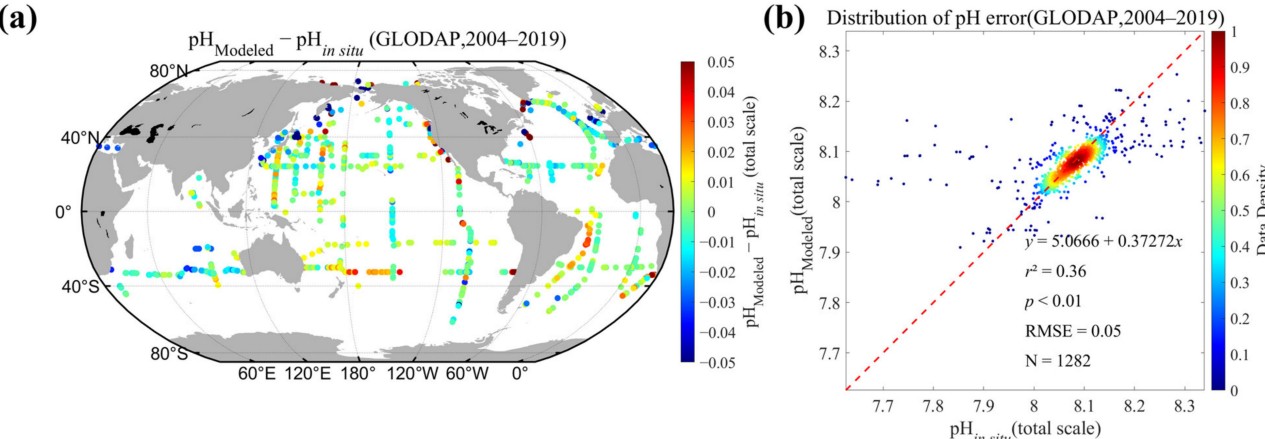

**Figure 13.** Independence validation based on the GLODAP *in situ* dataset. (**a**) Spatial distribution of the inversion error at the matchup validation points. (**b**) Point-to-point comparison between the modeled results and the *in situ* pH along the 1:1 line with $R^2 = 0.36$, RMSE = 0.050, and *n* = 1282. The red dotted line is 1:1 line.

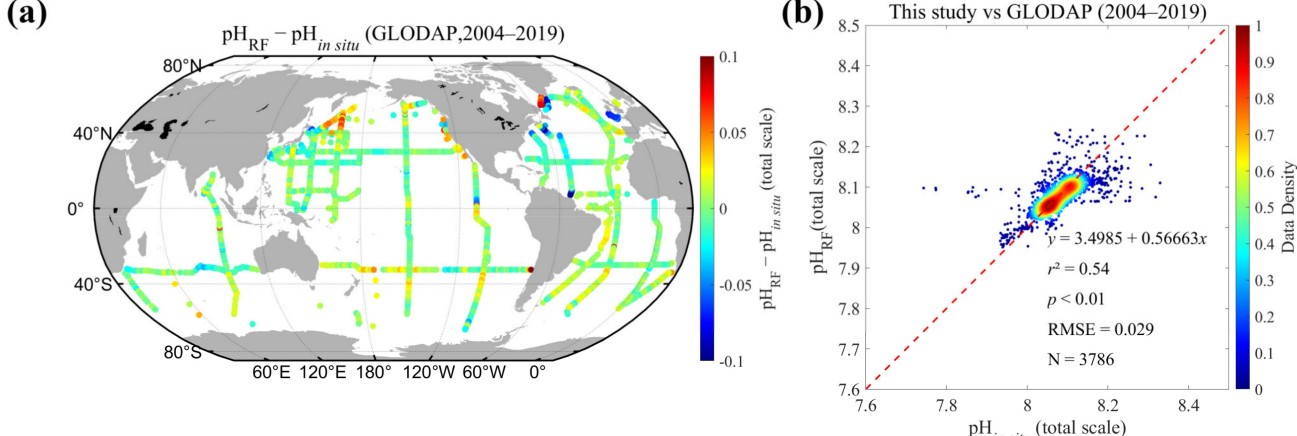

**Figure 14.** Global sea surface pH remote sensing product validation. (**a**) Sampling points after matching with the GLODAP *in situ* dataset, spanning the period 2004 to 2019. (**b**) Comparison between the *in situ* points and the estimated results on a 1:1 line, where $R^2 = 0.54$, RMSE = 0.029, and *n* = 3786.

For the global underway pH dataset and the sea surface pH remote sensing product, we selected the open ocean and marginal sea typical sea areas where the underway data are dense to calculate the sea surface pH variation rate in different sea areas, which are shown in Table 6. Specifically, for the underway data and remote sensing data, we calculated the monthly average value of the data in the selected grid points in the grid, and then calculated the pH variation rate by using the one-dimensional linear regression. The annual pH variation rate was then obtained by multiplying the rate of change by 12.

**Table 6.** Spatial extent of seven typical sea areas.

|   | Sea Area | Longitude | Latitude |
|---|---|---|---|
| 1 | Northwest Pacific Ocean | 141.5–146.5°E | 36–40°N |
| 2 | South of Australia | 142.5–147.5°E | 44–48°S |
| 3 | Equatorial Pacific Ocean | 150.5–155.5°W | 4.5–8.5°S |
| 4 | South of South America | 60.5–65.5°W | 55–59°S |
| 5 | North of Puerto Rico | 63.5–68.5°W | 18–22°N |
| 6 | Mid North Atlantic Ocean | 23.5–28.5°W | 36–40°N |
| 7 | Northeast Atlantic Ocean | 6.5–11.5°W | 46–50°N |

We plotted the data density distribution map based on the calculated underway pH (*in situ\**) data (Figure 15) and selected seven typical sea areas (4° (latitude) × 5° (longitude)) with enough data points for the long-time-series analysis, which are shown in Table 6. We also compared pH (*in situ\**) with time-series monthly averaged satellite-derived pH in the same grid (Figure 16). The satellite-derived pH was generally in good agreement with pH (*in situ\**), which further demonstrates the retrieval ability of our pH model. In the sea areas with dense pH (*in situ\**) values in Figure 15, we calculated variation trends based on the monthly pH from 2004–2019 by using one-dimensional linear regression to obtain the variation rate and then multiplying the rate by 12 to obtain the annual rate of change.

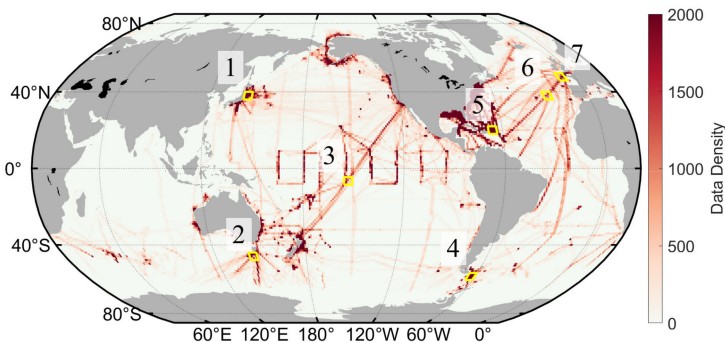

**Figure 15.** Density map of the calculated pH (*in situ\**) data from 2004 to 2019, with a grid point size of 1° × 1°. The seven numbers represent the seven sea areas in Table 6. The seven yellow boxes are 4° (latitude) × 5° (longitude) grids for statistical analysis.

Point 1 in the coastal region of the northwest Pacific Ocean, point 4 in the Drake Strait (the southernmost tip of South America), and point 5 to the north of Puerto Rico all had the greatest density of data and clearly showed significant pH decreasing trends, especially points 4 and 5, where the rates of decrease were almost $0.72 \times 10^{-4}$ and $0.49 \times 10^{-4}$ year$^{-1}$, respectively, over the past 16 years. Point 3, located south of the equatorial Pacific Ocean, also exhibited a significant decreasing trend in pH based on its available pH (*in situ\**) data during the period 2004–2016. In contrast to other points, point 3 exhibited relatively irregular interannual and seasonal changes, which may be due to the larger El Niño–Southern Oscillation event in this area.

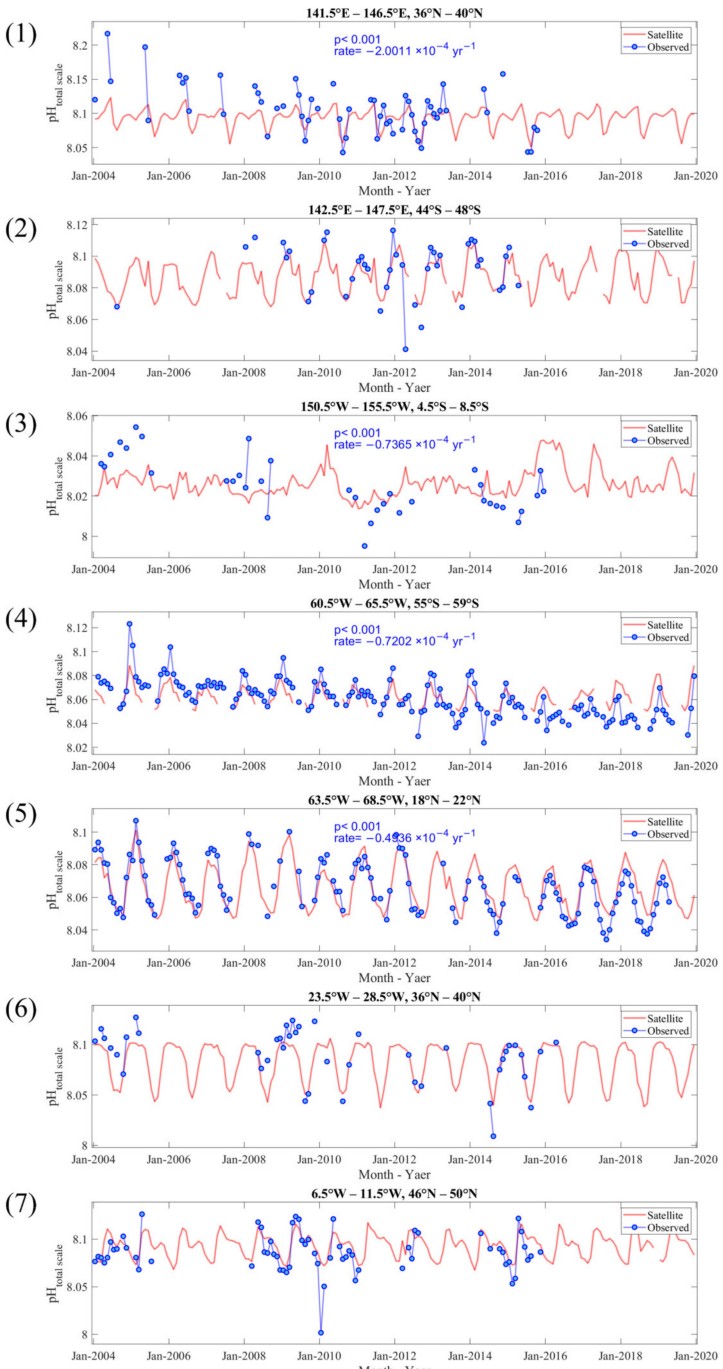

**Figure 16.** Seasonal variation of sea surface pH in seven typical regions with locations marked in Figure 15 and noted in Table 6: (**1**) northwest Pacific Ocean, (**2**) south of Australia, (**3**) equatorial Pacific Ocean, (**4**) south of South America, (**5**) north of Puerto Rico, (**6**) mid North Atlantic Ocean, and (**7**) northeast Atlantic Ocean. The red lines are the satellite-derived pH. The blue dot is the pH (*in situ\**); when the $p < 0.1$, the rate of change is marked in blue text. All points are the results of monthly averaging under a $4° \times 5°$ grid.

## 4. Results and Discussion

### 4.1. Comparison with Other pH Products

We filtered sea surface *in situ* pH data above 5 m water depth from the GLODAP dataset from 2004 to 2019 by Lueker's calculation criteria (SSS: 19–43 psu; SST: 2–35 °C). The pH values from GLODAP were then matched to each product and compared for error.

To ensure a fair comparison, the products in this study were compared at a reduced spatial resolution of $1° \times 1°$, while only the common valid points of all products were retained in the comparison.

It can be seen from Figure 17 that the accuracy of the JMA product is comparable to the product in this study, while the CMEMS product is more accurate. The pH products of JMA were calculated indirectly from TA and DIC obtained by multivariate regression in sub-sea. The pH products of CMEMS were calculated indirectly from $pCO_2$ reconstructed by the FFNN model and TA reconstructed by empirical formula. In this study, the global sea surface pH underway dataset is first reconstructed, and then the direct inversion is combined with remote sensing on the basis of this dataset. The method in this study provides another approach to obtaining seawater pH. Meanwhile, compared with the $1° \times 1°$ spatial resolution of JMA and CMEMS products, the product of this study has a higher spatial resolution ($0.25° \times 0.25°$). Because it uses MODIS's SST and Chla products as input, it will be conditioned to expand to 4 km spatial resolution in the future. Therefore, the model of this study has a higher extension value.

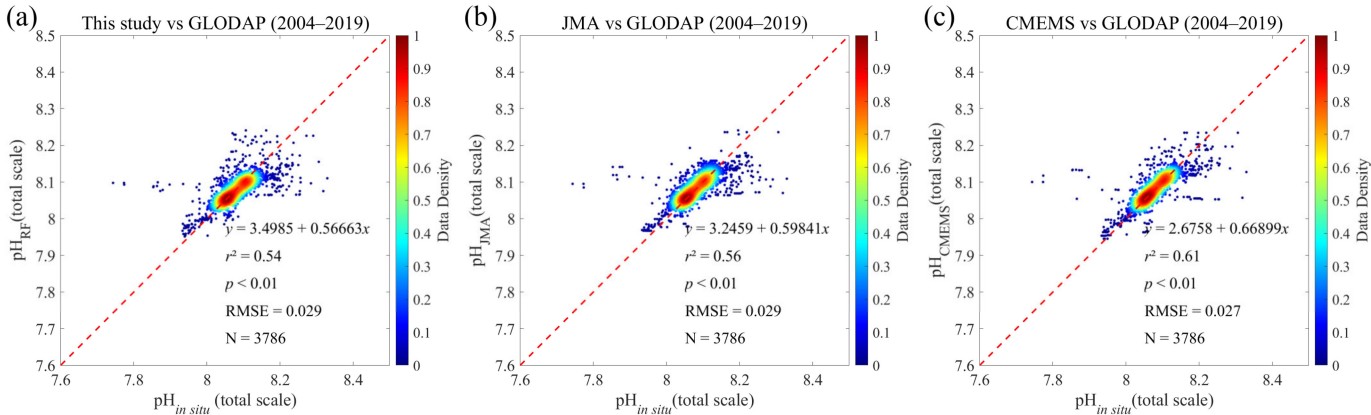

**Figure 17.** Scatter plot between satellite-derived pH and the *in situ* pH: (**a**) is pH product in this study, (**b**) is pH product from JMA, (**c**) is pH product from CMEMS.

### 4.2. Spatial Distribution of Sea Surface pH

As we had generated a massive scale of calculated pH (*in situ\**), we first present its climatologic average over the period 2004–2019 (Figure 18). The calculated pH (*in situ\**) values were averaged in a $1° \times 1°$ grid. In general, sea surface pH distribution is strongly consistent with the latitudinal gradient. Sea surface pH exhibited low values at low latitudes and high values at middle and high latitudes. Specifically, a very low pH was found in the eastern equatorial upwelling sea area of the Pacific Ocean, where the pH remained <8.0 all year round, which was affected by the deep and high concentration of $CO_2$ seawater. In the North Atlantic Ocean, the pH of the waters above 40° latitude was ~8.1 all year round. The pH in the North Indian Ocean waters was ~8–8.05, while the waters near the Middle East can even reach pH values of <8.0. Most of the Southern Ocean pH was ~8.05–8.1.

Based on the satellite-derived monthly averaged global sea surface pH (Figure 19), full pH data coverage was obtained. As the SST was one of the predictors, pH distribution showed significant latitudinal gradient and seasonal variation. High pH values were concentrated in the North Pacific Ocean from December to April. While the high-pH area gradually decreased after entering the summer with the increase in seawater temperature, the low-pH zone starts to shift northward and gradually squeezes the space of the high-pH zone. This phenomenon was also found in the South Pacific Ocean. Although the high-pH area in the South Pacific Ocean at 40° S from June to October was mainly concentrated to the east of Australia, there was a more continuous distribution of high values in the east–west direction. During the southern hemisphere summer, the east–west distribution

was no longer continuous, and the pH in the mid-latitude eastern part of the South Pacific began to decline, and the area of the high-pH area in the western part also shrank.

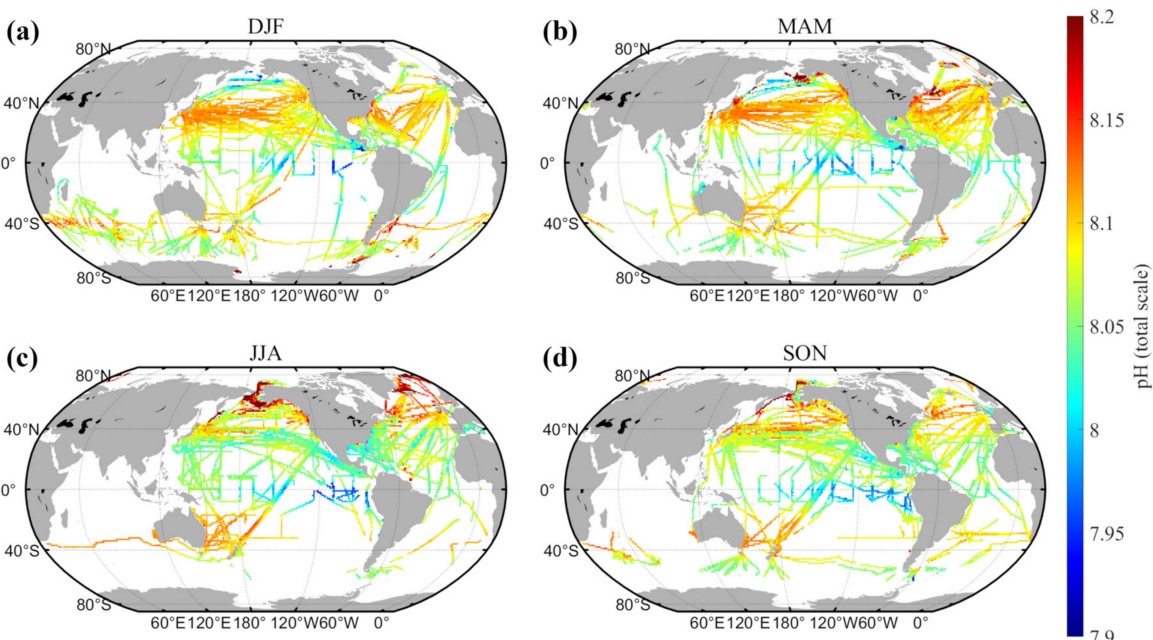

**Figure 18.** Global spatial distribution of seasonal sea surface pH. The letters at the top of each panel represent the initials of the months of the year: (**a**) for December to February, (**b**) for March to May, (**c**) for June to August, and (**d**) for September to November.

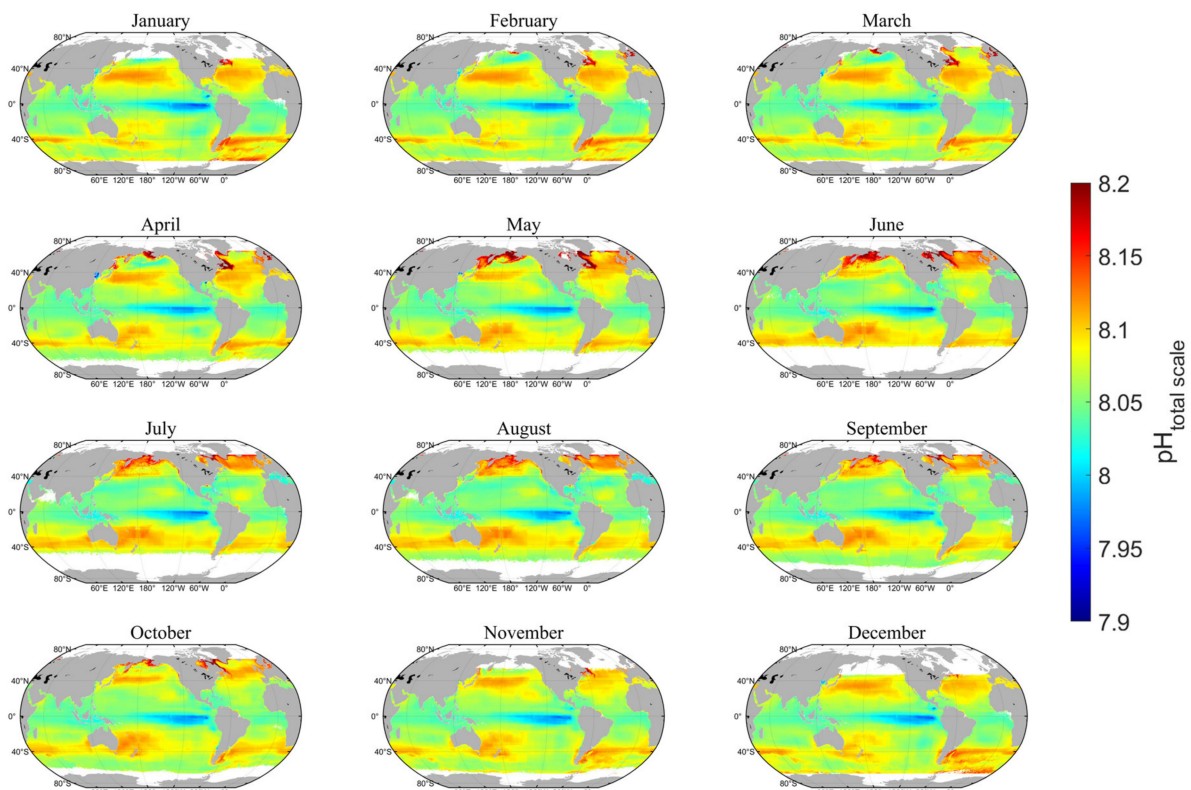

**Figure 19.** Climatological monthly averaged distribution of global sea surface satellite-derived pH.

## 5. Conclusions

Accurate estimation of sea surface pH by satellite remote sensing has been a challenging task, and it is difficult to simulate mechanistically and to construct models using traditional empirical methods. There is a lack of sufficient spatially and temporally representative datasets for model training. In this study, we developed a method to obtain a large-volume global sea surface pH dataset based on the LDEO underway $p$CO$_2$ dataset and estimated TA. On this basis, 18 experiments were set to compare three machine learning methods and eleven input parameter configurations. Finally, the RF-model-based sea surface pH remote sensing inversion model with LON, LAT, SST, Chla, and MLD as inputs that reflect spatial differences and physical and biological effects on sea surface pH was established, and it achieved satisfactory results based on various validation and sensitivity tests. Based on the model, we produced a monthly averaged product of sea surface pH at a spatial resolution of $0.25° \times 0.25°$ from 2004 to 2019. Meanwhile, the pH product of this study is at the same accuracy level as similar international products.

With the constructed calculated pH (*in situ*\*), we could clearly see the decreasing trend in pH in many sea areas globally. Time-series satellite-derived pH values also exhibited similar trends that were generally consistent with pH (*in situ*\*). These time-series datasets provide a refined view of the global pH variation on a high spatial and temporal scale, which will be of great help to global climate change studies and ecological environments.

The pH algorithm we developed was the direct inversion of pH. Although the pH inversion method in its current version was developed successfully and yields good results, there are still some unresolved problems in the pH calculation process: The original $p$CO$_2$ underway data had to be quality controlled twice by the publisher and the authors, and there is still an inevitable measurement error, which is also passed to the pH dataset along with the pH calculation. In addition, because the carbonate dissociation constants of Lueker [34] were used in the pH calculation, the data with SST of <2 °C were filtered in the calculation, resulting in a certain degree of missing pH data in high-latitude and polar areas. Therefore, the performance of this inversion model in these areas still needs to be optimized. A deeper investigation and parameterization should be conducted to improve the inversion accuracy. In this study, we only present the basic information of the global pH distribution and trends to show the performance of our algorithms and time-series satellite-derived pH dataset. More understanding of pH variation could be revealed based on these time-series datasets with detailed analysis of pH spatiotemporal distribution and inter-regional comparisons, which can also facilitate new parameterizations and algorithm updates.

**Author Contributions:** Conceptualization, Y.B. and X.H.; data curation, F.G.; methodology, Z.J. and Z.S.; resources, F.G.; software, S.Y.; validation, Z.J.; visualization, Z.J., S.Y. and S.Z.; writing—original draft, Z.J.; writing—review and editing, Y.B. and X.H. All authors have read and agreed to the published version of the manuscript.

**Funding:** This study was supported by the National Key Research and Development Program of China (Grant #2017YFA0603003), Key Special Project for Introduced Talents Team of Southern Marine Science and Engineering Guangdong Laboratory (Guangzhou) (GML2019ZD0602), the National Natural Science Foundation of China (Grants #42176177 and #41825014), and Zhejiang Provincial Natural Science Foundation of China (2017R52001 and LR18D060001).

**Data Availability Statement:** The data supporting reported results can be found in the Marine Satellite Data Online Analysis Platform (SatCO$_2$: https://www.satco2.com/).

**Acknowledgments:** We thank the SOED/SIO/MNR satellite ground station, satellite data processing and sharing center, and the marine satellite data online analysis platform (SatCO$_2$) for their help with data collection and processing. We thank LDEO and the National Centers for Environmental Information (NCEI)/National Oceanic and Atmospheric Administration (NOAA) (https://www.ncei.noaa.gov/access/ocean-carbon-data-system/oceans/LDEO_Underway_Database/. Accessed on 29 November 2020) for providing Global Surface $p$CO$_2$ (LDEO) Database V.2019. We thank Bjerknes Climate Data Centre and the ICOS Ocean Thematic Centre (https://www.glodap.info/. Accessed on 24 June 2020) for providing the GLODAP v.2.2020 dataset. We thank NASA's MODIS/Aqua

satellite (https://oceancolor.gsfc.nasa.gov/. Accessed on 17 November 2020) for providing the remote sensing reflectance, chlorophyll, and sea surface temperature data. We thank Remote Sensing Systems (www.remss.com. Accessed on 17 November 2020) for providing the CCMP Version-2.0 vector wind analyses dataset, and we thank Copernicus Marine Environment Monitoring Service (CMEMS) (https://resources.marine.copernicus.eu/. Accessed on 1 February 2022) for providing the MLD dataset.

**Conflicts of Interest:** The authors declare no conflict of interest.

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
