# Peer review of "Remote Sensing of Global Sea Surface pH Based on Massive Underway Data and Machine Learning"

_remotesensing, doi:10.3390/rs14102366_

Round 1

Reviewer 1 Report

I think the manuscript has been improved a lot an could be considered for publication in this journal.

Author Response

Response to Reviewer 1 Comments

Point 1:

I think the manuscript has been improved a lot an could be considered for publication in this journal.

Response 1:

Thank you very much for your comment.

Reviewer 2 Report

On lines 28-29 the terms R2 and RMSE are used without first defining them.  This should be avoided.

Over all, the graphs, tables, and formulas are very well presented and appreciated.

I feel that this is a great step in learning and understanding the pH issues caused by CO2, and is a potentially great method for tracking it.  Clearly, the paper has been through some revision and other editors have put forth their opinions:  the authors have responded to these with a great deal of additional explanation, work, and science.  The ample references further enforce this.

Author Response

Response to Reviewer 2 Comments

Point 1:

On lines 28-29 the terms R2 and RMSE are used without first defining them.  This should be avoided.

Over all, the graphs, tables, and formulas are very well presented and appreciated.

I feel that this is a great step in learning and understanding the pH issues caused by CO2, and is a potentially great method for tracking it.  Clearly, the paper has been through some revision and other editors have put forth their opinions:  the authors have responded to these with a great deal of additional explanation, work, and science.  The ample references further enforce this.

Response 1:

Thank you for your comments. We have added the definitions at line 28 of the manuscript.

The full expression of this sentence is:

After several tests of machine learning methods and groups of input parameters, we chose the random forest model with longitude, latitude, sea surface temperature (SST), chlorophyll a (Chla) and Mixed layer depth (MLD) as model inputs with the best performance of correlation coefficient (R² = 0.96) and root mean squared error (RMSE = 0.008) in the training set and R² = 0.83 (RMSE = 0.017) in the testing set.
